# 3D printing of multi-scalable structures via high penetration near-infrared photopolymerization

Junzhe Zhu [1,2], Qiang Zhang[3], Tianqing Yang [1,2], Yu Liu[3,4 ✉] & Ren Liu[1,2 ✉]

3D printing consisted of in-situ UV-curing module can build complex 3D structures, in which direct ink writing can handle versatile materials. However, UV-based direct ink writing (DIW) is facing a trade-off between required curing intensity and effectiveness range, and it cannot implement multiscale parallelization at ease. We overcome these difficulties by ink design and introducing near-infrared (NIR) laser assisted module, and this increases the scalability of direct ink writing to solidify the deposited filament with diameter up to 4 mm, which is much beyond any of existing UV-assisted DIW. The NIR effectiveness range can expand to tens of centimeters and deliver the embedded writing capability. We also demonstrate its parallel manufacturing capability for simultaneous curing of multi-color filaments and freestanding objects. The strategy owns further advantages to be integrated with other types of ink-based 3D printing technologies for extensive applications.

[1] Key Laboratory of Synthetic and Biological Colloids, Ministry of Education, School of Chemical and Material Engineering, Jiangnan University, 214122 Wuxi, Jiangsu, China. [2] International Research Center for Photoresponsive Molecules and Materials, Jiangnan University, 214122 Wuxi, Jiangsu, China. [3] School of Mechanical Engineering, Jiangnan University, 214122 Wuxi, Jiangsu, China. [4] Jiangsu Key Lab of Advanced Food Manufacturing Equipment and Technology, Jiangnan University, 214122 Wuxi, China. ✉email: yuliu@jiangnan.edu.cn; liuren@jiangnan.edu.cn

n a traditional Chinese myth of "Shenbi Ma Liang", the boy called Ma Liang got a magic paintbrush toward which he could momentarily create real things. Manufacturing technologies are always concerned by human beings for characteristics of fast speed and flexibility, and the progress in civilization on technology is always accompanied with applying of methodologies and materials[1–4]. Lately, 3D printing has quickly grown and attracted a significant amount of attentions in biomedicals[5], origami[6], data matter[7], microfluids[8], microelectronics[9,10], etc. Direct ink writing (DIW), as an extrusion-based 3D printing (3DP), can incessantly stack inks, thus fabricating objects rapidly and freely[1,11], which is similar to the paintbrush of Ma Liang. Combining with different choices of inks for DIW, it is possible to fabricate numerous objects with unique features, for example, 4D printing[12], electromechanical properties[9,13], bioprocess intensification[14], porous materials[15], polymer foams[16], and shape memory[17]. However, purely thermal-curing inks have limited DIW to the attempt of implementing complex 3D geometry and functional structures, as the post treatment is essential and time-consuming[16,18]. Moreover, the size of uncured DIW-printed objects is also limited owing to the poor mechanical behavior.

Ink design is, predominantly, one of the greatest challenges in DIW[19]. Photopolymerizable inks, for its reasonable curing kinetics as well as mild facilities[20], have received revitalized attentions in a variety of 3D printing[2,3,21], such as stereolithography (SLA), direct laser writing (DLW), digital light processing (DLP), ink jetting, and real-time curing of DIW. Currently, ultraviolet (UV) and blue lights are the main irradiation sources employed to instantly trigger the photoreaction spatially and actinically, thus the cross-linking motif could be achieved with a target object obtained[22,23]. Despite the convenience offered by UV and blue lights, intensity of short wavelength lights is a trade-off for better mechano-properties by uniformed polymerization and health concern of UV exposure. Among the photopolymerization methodologies, near-infrared light (NIR) has a more salient role in rapid deep-curing for its remarkable penetration in various media by employing upconversion strategies[24–27]. Recently, we have applied NIR light in free radical as well as cationic polymerization by utilizing upconversion nanoparticle (UCNP), and successfully photocured resins for over 10 cm with relatively uniformed conversion[28–30], which is promising to improve the curing manner and valuable to additive manufacturing.

In the presented literature, we report a 3DP strategy with NIR-induced photopolymerization, and the fusion of NIR photocurable material and DIW 3DP technology could achieve in-situ curing of thick filament with high penetration. Toward this methodology, we obtain multi-scalable filaments and realize fabrication in different color as well as multi-colors. Moreover, several freestanding objects are roughly printed by this method.

## Results

**NIR-DIW ink preparation and real-time FTIR photorheology analysis.** Differentiated materials could signature, slightly or significantly, different natures of the essential materials, thus promising 3D-printed objects with prominent characteristics. For DIW printing, a well-printed structure is in the dominion of proper rheology properties of ink[19]. The inks should be gel-liked under storage while showing flowing behavior with shear force and eventually hold the shape at pre-programmed position to form a precise structure. Before we applied NIR-induced photopolymerization into DIW printing, a series of rheological analysis and real-time FTIR-rheology test[31] was executed to simulate the DIW procedure and optimize the method. In the previous study, we found that a UCNP concentration of 0.3 w.t.% could result

curing depth. Nevertheless, curing speed is an essential demand in 3DP process, thus we introduce a higher concentration of UCNP (1.0 w.t.%) and photoinitiator (1.0 w.t.%) for a more rapid polymerization. As shown in Fig. 1c, at the beginning of irradiation the vinyl bond conversion increased slowly, whereas the modulus exhibited a leap and with getting lower followed, which may be the effect directly lead by NIR heat thus thinning of ink dominating the properties. After gel point reached, the rheological property of cross-linked ink was no longer sensitive to the temperature and the modulus significantly increased under exposure. The conversion curve rose rapidly could be attributed to an accelerated rate since radical polymerization of acrylate is sensitive to temperature.

**3D printing by NIR-DIW.** As the critical dose point of NIR-DIW ink was found in kinetic and rheology tests above, we applied the NIR laser to a typical DIW setup, in which the laser beam was vertical to the nozzle and aligned with the point 2 mm below the nozzle. To confirm that the NIR light intensity would influence the extruded ink, a variety of different laser power was set to trigger the photopolymerization of DIW patterned (2.50 mm nozzle) lines on glass plate. The printed samples were grinded to measure the IR spectra by attenuated total reflection (ATR) FTIR spectroscopy to analyze the conversion after NIR-DIW. As shown in Fig. 2b, the conversion of vinyl group shows a positive correlation to light intensity and indicates that higher light intensity does promote the chemical reactions, which is an analogy to increasement of the dose in Fig. 1c. Spontaneously, we employed NIR-DIW to print wood pile structure (Fig. 2d), and DIW with after-printing NIR treatment was also performed (Fig. 2c) to show the part NIR light taken in shape-holding.

Based on the penetration of NIR light, six types of nozzles with diverse diameters (0.21, 0.41, 0.84, 1.55, 2.50, 4.00 mm) were used in order to explore the effect brought by width of filament. The extrusion speed and light intensity were all the same during line-patterning by NIR-DIW. From the point of view of ATR-FTIR spectroscopy as Fig. 3a, a similar curing ratio of different samples occurred with conversions at around a half even in the filament extruded by 4.00 mm nozzle, except the conversion of sample extruded by 0.21 mm nozzle could achieve 70.4%. And microscale printing was also attempted utilizing a 5 μm capillary nozzle, and 17 μm filament was successfully patterned as shown in Supplementary Fig. 4.

To evaluate the versatility of NIR-DIW in complicated ink composition containing light extinction ingredients, such as pigments in this case. As the absorptions of pigments cover the regions of UV–vis light employed in present 3D-printing methods based on photopolymerization, hardly could we obtain the colored structure through photocurable-ink-based 3D printing equipped with external light sources. Energy absorbing should be relied on the width of decay path in colored inks according to Lambert–Beer law, which indicated that photocuring strategies are limited to the applications with exogenously multicolor 3D printing.

## Discussion

In this study, we demonstrate how NIR photopolymerization gets integrated into DIW 3D printing. According to the result of real-time FTIR-rheology analysis, it is important for NIR-DIW to control the irradiating dose to ensure the ink gets across the gel point instead of the structures getting heated and softened as shown in Fig. 1c. And the structure cured by in-situ NIR photopolymerization (Fig. 2d) differed from the post-NIR photopolymerization structure (Fig. 2c), which indicates that it is the real-time curing induced by NIR light that endued the structure

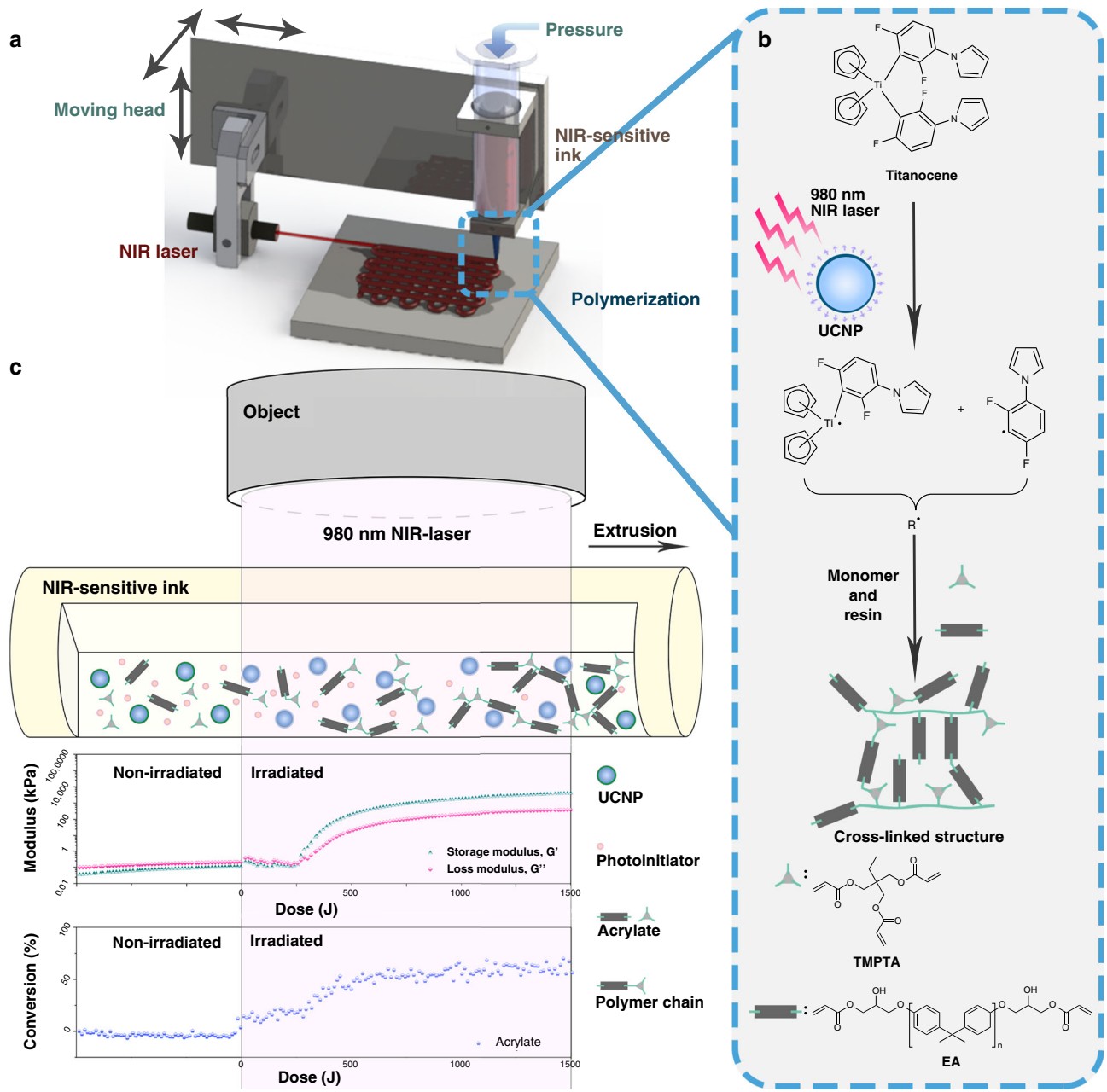

**Fig. 1 Schematic of NIR-DIW and real-time FTIR photorheology analysis. a** Scheme of NIR-induced DIW setup; **b** structures and reactions applied in NIR-DIW printing; **c** monitoring of NIR-induced photopolymerization by real-time FTIR rheological analysis.

with spatial resolution other than the rheology feature of the ink. As the conversions show little difference (Fig. 3a), the NIR-DIW is promising in thick filament printing to achieve larger printing scale while extra fine filament could be also patterned utilizing a 5 μm capillary nozzle (Supplementary Fig. 4). It may be possible that the time-consuming process of internal filling can be accelerated, whereas an external with high resolution could be promised by applying multiscale (from 17 μm to 4 mm) filaments.

As for NIR-induced photopolymerization, the longer-wavelength irradiation source is promising for a better penetration in such light-extinct photocuring system through an exogenously colored deep-curing filament with existence of different pigments by NIR curing[32]. We have, in present paper, applied this feature to envision NIR-DIW method, where a series of differentiated color lines were successfully printed. In Fig. 3c, we measured the absorption spectra of Titanocene and pigments as

well as the emission of UCNP, all the absorption of pigments used covered the major absorbing peak of titanocene which is commonly used in visible light 3D printing and also used in this research. When applying to NIR-DIW printing of lines, as Fig. 3b indicates, almost all the inks exhibited normal curing ratio at about 40% slightly lower than the sample stated above (Fig. 3a). Owing to the good penetration of NIR light, impressive different colored structures obtained and prove the versatility of NIR-DIW in exogenously multicolor 3D printing. Besides, a possibility of multicolored coaxial filament was printed by NIR-DIW as displayed in Fig. 4a, b, which presents a solution to obtain core-shell liked coaxial filaments with multi-material characteristic. By applying NIR-induced photocuring, it could be imagined to rapidly fabricate multi-material filament for a better performance thus promise the manufacturing of functionalized objects or programmed structures through this technique[10,23,33,34].

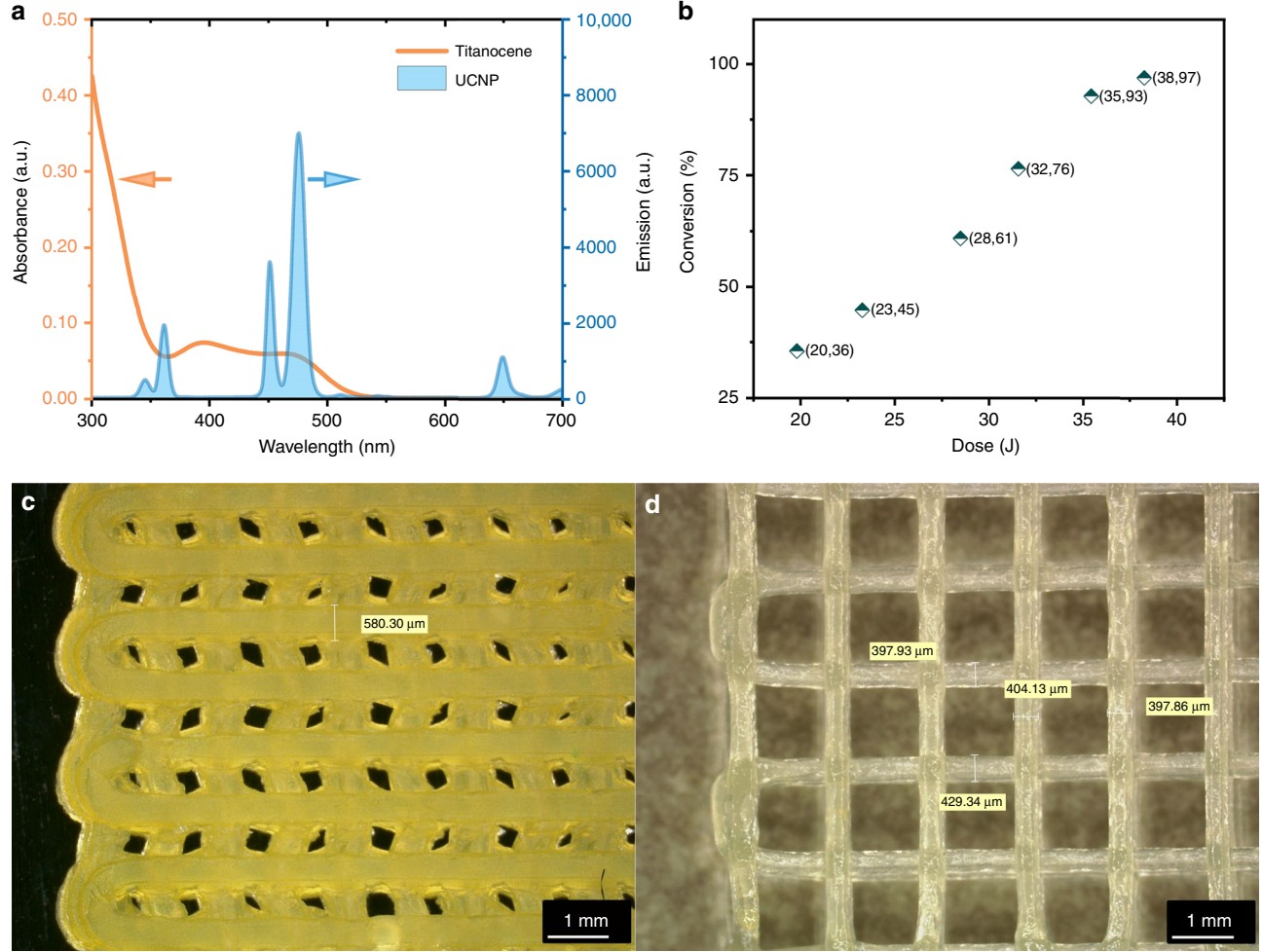

**Fig. 2 Matching of initiation system and differences between varied photopolymerization conditions. a** Absorption spectrum of titanocene and emission spectrum of UCNP; **b** the conversion of vinyl group vs. the irradiationdose as monitored by ATR-FTIR; optical images of the structures by **c** post-NIR irradiated or **d** in-situ NIR curing, the scale bar is equivalent to 1 mm.

Freestanding objects fabrication has drawn growing interests in additive manufacturing for the combination of features, which ensure the human beings of freely create objects like the story of "Shenbi Ma Liang" in beginning. However, seldom do these freestanding structures get well prepared through DIW for the material properties currently could not retain the structure before thermal post-treatments. Several salient strategies have been employed by introducing laser heat[35] or frontal ring opening metathesis polymerization (FROMP)[36]. We tried to fabricate freestanding objects, and with NIR-DIW rapid curing the structure directly stacked to form freestanding spring or character "M"-shaped cantilever structure with different colors as exhibited in Fig. 4c, d, which may present a method to realize 3D printing of freestanding objects.

Furthermore, monolith 3D-printed structures could be fabricated by this method as well. As exogenously colored filaments could be real-time cured and patterned, we fabricated a series structures with different colors employing pigment-free ink or colored inks as shown in Fig. 5a, b, where the structure could be well repeated despite the pigments contained, whereas the printed objects showed similar robustness in tensile test (Fig. 5a) and the structures as well as the colors were retained under various conditions (heat, humidity, and acidic solution; Supplementary Fig. 5). And we also applied NIR-DIW in manufacture of similar structures with different resolutions, displayed in Fig. 5c, due to

the real-time curing of filaments extruded by multi-diameter nozzles.

As this fabrication methods could promise scientific and technical advances, DIW has drawn growing interests in the device-manufacturing in physical, biological, and chemical applications based on thermoset or photocurable inks. The work of NIR-DIW we presented is aiming to provide a 3D-printing method. In this case, the application of UCNP in-situ light realized curing of multi-diameter nozzle extruded filaments as well as different colors. Thanks to the penetration of NIR light, both the pigment-free or colored filaments could achieve similar conversions without disturbing by pigments absorption. Moreover, multicolor filament was printed while freestanding objects were also fabricated, which may present solution of multi-material and freestanding objects fabricating. We are sure this work of NIR-DIW methodology would be of tremendous use to further rapid manufacturing of smart materials with multi-components, excellent objects as well as gigantic objects where thick filaments could fast fill the inner part while fine filaments could promise a meticulous surface.

## Methods

**Materials**. Photoinitiator titanocene (bis(cyclopentadienyl)bis[2,6-difluoro-3-(1-pyrryl)phenyl]titanium) was obtained from BASF (China) Co., Ltd. Difunctional bisphenol A epoxy acrylate oligomer (EA) RY1102A80 and trimethylolpropane

**Fig. 3 Patterned filaments utilizing different nozzles and colored inks.** The conversion of vinyl group vs. **a** nozzle diameter or **b** colors as monitored by ATR-FTIR, and error bars are reported as standard error of mean, and the scale bar is equivalent to 2 mm; **c** absorption spectra of titanocene and pigments used in different formula.

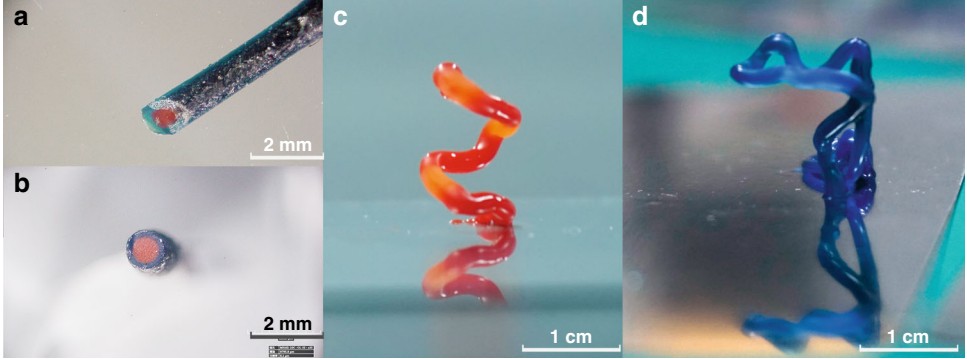

**Fig. 4 NIR-DIW-printed multicolor filament and colored freestanding structures. a** Top view and **b** side view of core-shell multicolor filament, the scale bar is equivalent to 2 mm; **c** freestanding spiral structure with red pigments; **d** freestanding M-shaped cantilever structure with blue pigments, the scale bar is equivalent to 1 cm.

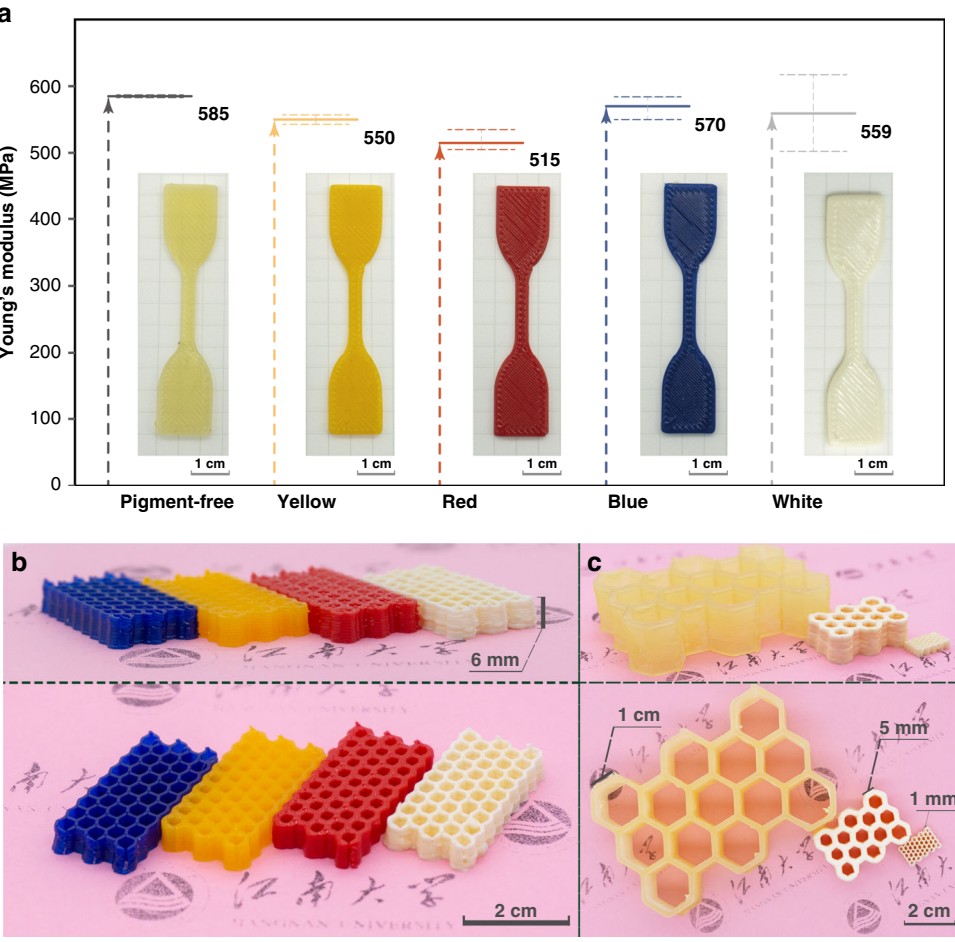

**Fig. 5 NIR-DIW-printed objects with different colors and resolution. a** Young's modulus obtained by tensile test of 3D-printed samples, and error bars are reported as standard error of mean, and the scale bar is equivalent to 1 cm; **b** side view and top view of same structure printed by different pigments; **c** side view and top view of 3D-printed monolith structures in different resolution, the scale bar is equivalent to 2 cm.

acrylate (TMPTA) were supplied by Jiangsu Kailin Ruiyang Chemical (China) Co., Ltd. Silicon dioxide Ace Matt TS100 was a product of Evonik (China) Co. Ltd. Yellow, red, and blue were provided by Jiangsu Kuangshun Photosensitivity New-material Stock Co. White pigment was a product of DuPont (China) Co., Ltd. UCNP (NaYF4:Yb,Tm; mean diameter: 1385 nm; polydispersity: 0.396; TEM image attached in Supplementary Fig. 1) was purchased from Shanghai Ziqi Chemical Technology Co., Ltd. The NIR-DIW ink was prepared by mechanically mixing the mixture of titanocene (1.0 w.t.%), UCNP (1.0 w.t.%), TS100 (13.0 w.t.%), EA (42.5 w.t.%), and TMPTA (42.5 w.t.%). For colored inks, the composition contained pigment (yellow, red, blue, black, or white; 0.5 w.t.%) titanocene (1.0 w.t.%), UCNP (1.0 w.t.%), TS100 (12.5 w.t.%), EA (42.5 w.t.%), and TMPTA (42.5 w.t.%). All the inks were defoamed by centrifugation before using.

**3D-printing methods**. All of the 3D-printing experiments were carried on a home-built DIW 3D printer, consisting of a computer-controlled 3-axis gantry platform, a high-pressure booster, micro nozzles with various diameters, and a newly proposed NIR (FC-W-980H-50W, Changchun New Industries Optoelectronics Technology Co.,Ltd.) curing module. Desired architectures were prepared through controlled extrusion of the ink onto glass or steel substrate, accompanying with controlled moves along the $X$ and $Y$ axes. Printing paths were compiled as parameterized G-code scripts, and designed to maximize continuity within each printed layer. A coaxial ink extrusion module is equipped with a coaxial extruder, shell material reservoir, core material reservoir, and two separate ink supply modules as shown in Supplementary Fig. 2.

For preparation of core-shell tubing, the coaxial extruder was made of stainless steel and its detailed dimensioned are 1.3 mm (outer layer diameter) and 0.5 mm (inner layer diameter), which was fixed through the whole study. Pneumatic dispensers were modified for handling with high ink viscosity with high-pressure resolution 1 kPa. All movements were programmed in LabVIEW.

**Characterization**. UV–Vis absorption spectra of titanocene and pigments were dispersed in acetonitrile employing a spectrophotometer model as EMCLAB EMC-61PC-UV. Tensile test was exhibited by Instron 5967X Universal Testing Systems.

Conversion of functional groups were monitored by computing the decrease of characteristic absorption peak utilizing ATR-FTIR. Real-time FTIR rheological analysis was performed with NIR light irradiation employing a setup fused with ATR-FTIR (Nicolet iS10 series, ThermoFisher) and rheometer (HAAKE MARS60 equipped with Rheonaut annex, ThermoFisher), as shown in Supplementary Fig. 3. The conversion of functional groups ($K_{f,t}$) was calculated as followed equation:

$$K_{f,t} = \left(1 - \frac{p_{f,t}/p_{f,i}}{p_{r,t}/p_{r,i}}\right) \times 100\%, \tag{1}$$

where $P_{f,t}$ and $P_{f,i}$ stands for integrated peak area of functional groups at different status or initial status, and $P_{r,t}$ and $P_{r,i}$ are integrated reference peak area of non-reactive groups at different status or initial status to exclude interference of physical factors.

## Data availability

Raw data were monitored by ThermoFisher software suits for HAAKE rheometer and Nicolet FTIR. Derived data supporting the investments of this 3D-printing method require computational processing by ThermoFisher exclusive data analysis suits and are therefore available from the corresponding author upon request. The raw data of UV–vis absorption spectra and tensile tests are available from the corresponding author upon request.

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

## Acknowledgements

We acknowledge the financial support by the National Nature Science Foundation of China (51673086, 51875253), National First-class Discipline Program of Food Science and Technology (JUFSTR20180301), and MOE & SAFEA for the 111 Project (B13025).

## Author contributions

R.L. and Y.L. conceived the project; J.Z., Y.L., and R.L. designed the experiments; J.Z. conducted all experiments; J.Z., Q.Z., and Y.L. developed the 3D-printing method; J.Z. and T.Y. analyzed and interpreted the results; J.Z., Y.L., and R.L. wrote the manuscript, with significant contributions from all authors. All authors approved the final version of the manuscript.

## Competing interests

The authors declare no competing interests.
