## [Peer Review File · Nature Communications]

REVIEWER COMMENTS

Reviewer #1 (Remarks to the Author):

This is a very interesting manuscript. The authors disclose 3D printing of free standing objects based on an up-conversion protocol applying NIR laser radiation at 980 nm for excitation of UCNPs. They also applied different colors for printing for which regular UV printing would fail.

This is from my point of view a difficult objective. I had for myself similar ideas many months ago but I failed.

Authors designed a special ink that facilitates manufacture of the aforementioned targets. UCNPs generate only 1 % UV photons accessible to initiate UV processes applying NIR laser radiation at 980 nm. The remaining part is heat which is from my point of view responsible for the observed reactions disclosed in Figure 1. Although the material started to polymerize, the overall elasticity of the material is high enough for processing while higher higher conversions provide the necessary mechanical properties as shown at later by increase of the respective modulus.

This paper combines a clever combination of chemistry and engineering which results only in a combination well functioning 3D printing application described in this manuscript. It is from my point novel and presumable saves processing time compared to two-photon excitation. I am convinced this paper will have a deep impact to the community. I would also expect further industrial applications based on the knowledge disclosed I this manuscript. Typical applications may cover areas in medicine to fabricate prosthesis, mechanical engineering to make prototypes for lab testing - just to name o few of them.

The authors also concluded well prospective developments for the future.

There are only a few small changes necessary. Ref. 27 does not relate to up-conversion. An alternative would be

D. Oprych, C. Schmitz, C. Ley, X. Allonas, E. Ermilov, R. Erdmann, B. Strehmel, "Photophysics of Up-Conversion Nanoparticles: Radical Photopolymerization of Multifunctional Methacrylates Comprising Blue- and UV-Sensitive Photoinitiators" ChemPhotoChem 2019, 3, 1119-1126.

fitting much better since it also explains the right up-conversion process. Furthermore, the labels in the figures are too small, they should be enlarged and spelling errors in the figures must be corrected. Neevrtheless, this is a small thing which can be easily fixed. Again, it is from my point of view a very interesting paper and I enjoyed to read it. I learned many things from it.

Authors should also better comment on the processing time to fabricate such objects. Desirable would would be to make a rough comparison with two-photon excitation since this represents a competitive technique. The advance of this manuscript is, although it also bases on multi-photon excitation, the fact that this happens in serial steps shall lead to shorter processing cycles. This method is from my point of view the more efficient way to fabricate the objects disclosed.

Experiments were disclosed somehow that experts in the field shall be able to rework and reproduce the results disclosed by the authors.

I strongly recommend publication of this ms.

Reviewer #2 (Remarks to the Author):

The authors in this paper suggest the use of NIR in 3D printing applications. The starting point is

their previous studies in the use of NIR light in free radical as well as cationic polymerization by utilizing up-conversion nanoparticle (UCNP), and successfully photocured resins for over 10 cm with relatively uniformed conversion. In this paper, the authors report a novel 3DP strategy with NIR induced photopolymerization, and the fusion of NIR photocurable material and DIW 3DP technology. By this strategy they could achieve in-situ curing of thick filament with high penetration. The paper is very well written and all the curing parameters well investigated. Besides designing the photocurable formulations and the irradiation conditions, the authors fabricate freestanding objects, and with NIR-DIW rapid curing the structure directly stacked to form freestanding spring with different colours. This is definitely very interesting in the field of 3D-printing and it is an important step forward in finding new technologies for 3D printing. For this reason I would suggest to accept this paper in Nature Communication in the present form without any modifications.

Reviewer #3 (Remarks to the Author):

Zhu et al. presented the fabrication of three-dimensional architecture enabled by up-conversion nanoparticles (UCNPs), which utilize near-infrared light for the generation of free radicals to initiate the radical polymerization of acrylate monomers and crosslinkers. The introduction of UCNPs greatly reduces the energy required for the crosslinking of a network and increases the light penetration length for a more holistic structure. The authors demonstrated preliminary results in the manuscript, showing interesting aspects of introducing UCNPs into 3D printing. However, early work has shown that up-conversion materials could be employed in stereolithography for building up three-dimensional objects (J. Méndez-Ramos, et al., J. Mater. Chem. C, 2016, 4, 801–806) and it would be advised for the authors to further clarify the design novelty compared with aforementioned work to distinguish a pure adaptation to DIW. Hence, this paper is not novel enough to warrant publication in Nat. Commun. And the reviewer would like to suggest the authors submit this paper somewhere else.

1. The manuscript is entitled "3D Printing of Multi-scalable Structures via High Penetration Near-Infrared Photopolymerization" but the concept of "multi-scale" was not addressed in the manuscript, nor did this reviewer trace experiments relating to multi-scale structure. It would be better if the authors would clarify or revise accordingly. This reviewer would like to suggest the authors listing the scales of existing UV-assisted DIW for the purpose to compare.
2. The schematics in Figure 1 were incorrect topologically since acrylate is not monovalent and the authors should revise this.
3. The conversion of the vinyl groups was calculated using ATR-FTIR. However, it has been known that IR spectra is more a qualitative rather than a quantitative analysis. This reviewer is concerned about how accurate the conversion obtained by the authors is since all the conversions are presented with three significant numbers.
4. The UCNPs in the manuscript, even though promising in near-infrared polymerization, were not provided with any chemical structure, such as the chemical composition, doping elements, size, PDI, and so on. It would be better if the authors could further clarify all the essential information.
5. The freestanding 3D-printed objects in Figure 4, in the opinions of this reviewer, are far from controllable if not intentionally designed that way. This reviewer is concerned about how precise this technique developed by the authors could be if the ink cannot be solidified. To demonstrate the 3D printability, more 3D printed monolith should be demonstrated other than spiral structure.
6. There are numerous grammatical errors. The authors are suggested to proofread more carefully to polish their manuscript significantly. Some (incomplete) examples are shown below.

In line 21, "facing with" should be "facing"

In line 36, "progresses" should be "progress"

In line 45, "has" should be "have"

In line 78, "a series works of" should be "a series of"

In line 92, "get" should be "gets"

In line 99, "glass palate" should be "glass plate"

In line 326, "nozzze" should be "nozzle"

7. From this paper, it is unclear to this reviewer how the pigments were integrated into the system and whether the printer colors were robust against different conditions.

8. This reviewer suggests some mechanical measurements on the printed structures to exhibit the robustness of the printed objects.

>**Reviewer 1:** This is a very interesting manuscript. The authors disclose 3D printing of free standing objects based on an up-conversion protocol applying NIR laser radiation at 980 nm for excitation of UCNPs. They also applied different colors for printing for which regular UV printing would fail.

This is from my point of view a difficult objective. I had for myself similar ideas many months ago but I failed.

Authors designed a special ink that facilitates manufacture of the aforementioned targets. UCNPs generate only 1 % UV photons accessible to initiate UV processes applying NIR laser radiation at 980 nm. The remaining part is heat which is from my point of view responsible for the observed reactions disclosed in Figure 1. Although the material started to polymerize, the overall elasticity of the material is high enough for processing while higher higher conversions provide the necessary mechanical properties as shown at later by increase of the respective modulus.

This paper combines a clever combination of chemistry and engineering which results only in a combination well functioning 3D printing application described in this manuscript. It is from my point novel and presumable saves processing time compared to two-photon excitation. I am convinced this paper will have a deep impact to the community. I would also expect further industrial applications based on the knowledge disclosed I this manuscript. Typical applications may cover areas in medicine to fabricate prosthesis, mechanical engineering to make prototypes for lab testing - just to name o few of them.

The authors also concluded well prospective developments for the future.

This reviewer's comments did help to improve the quality of the mechanism as well as the overall manuscript. Thanks to this reviewer who spent valuable time to work on the manuscript.

>1: There are only a few small changes necessary. Ref. 27 does not relate to up-conversion. An alternative would be

D. Oprych, C. Schmitz, C. Ley, X. Allonas, E. Ermilov, R. Erdmann, B. Strehmel, "Photophysics of Up-Conversion Nanoparticles: Radical Photopolymerization of Multifunctional Methacrylates Comprising Blue- and UV-Sensitive Photoinitiators" ChemPhotoChem 2019, 3, 1119-1126.

fitting much better since it also explains the right up-conversion process.

Thank you to the professional comment on the reference quoted about the up-conversion process. We have replaced Ref. 27 with the one "Photophysics of Up-Conversion Nanoparticles: Radical Photopolymerization of Multifunctional Methacrylates Comprising Blue- and UV-Sensitive Photoinitiators" suggested in the comment.

>2: Furthermore, the labels in the figures are too small, they should be enlarged and spelling errors in the figures must be corrected. Nevertheless, this is a small thing which can be easily fixed. Again, it is from my point of view a very interesting paper and I enjoyed to read it. I learned many things from it.

We resized the labels in the figures and corrected the spelling errors. Thanks again for spending time pointing out the mistakes and unthoughtful typesetting in the manuscript.

>3: Authors should also better comment on the processing time to fabricate such objects. Desirable would be to make a rough comparison with two-photon excitation since this represents a competitive technique. The advance of this manuscript is, although it also bases on multi-photon excitation, the fact that this happens in serial steps shall lead to shorter processing cycles. This method is from my point of view the more efficient way to fabricate the objects disclosed.

Experiments were disclosed somehow that experts in the field shall be able to rework and reproduce the results disclosed by the authors.

I strongly recommend publication of this ms.

This is absolutely a good idea to comment on the processing time to fabricate 3D printed monolith structure. We tried to print a cubic wood-pile structure to evaluate the time consumed in processing, which was made of 10.0 x 10.0 x 10.0 mm with 2.0 mm line width and 0.5 mm layer height adding up to 1,200 mm printing path in the whole printing. In the presented method, the processing time took about 4 min in which the printing speed could reach 6.0 mm s⁻¹, while the processing time for such structure may take several hours in two-photon

polymerization (2PP) method. Nevertheless, 2PP is an impressive additive manufacturing method for its superb resolution in micro-and nano- scale to enrich the innovation in micro-scale for humanity. We also added some 3D printed object and provided more details in the revision.

>**Reviewer 2:** The authors in this paper suggest the use of NIR in 3D printing applications. The starting point is their previous studies in the use of NIR light in free radical as well as cationic polymerization by utilizing up-conversion nanoparticle (UCNP), and successfully photocured resins for over 10 cm with relatively uniform conversion. In this paper, the authors report a novel 3DP strategy with NIR induced photopolymerization, and the fusion of NIR photocurable material and DIW 3DP technology. By this strategy they could achieve in-situ curing of thick filament with high penetration.

The paper is very well written and all the curing parameters well investigated. Besides designing the photocurable formulations and the irradiation conditions, the authors fabricate freestanding objects, and with NIR-DIW rapid curing the structure directly stacked to form freestanding spring with different colours.

This is definitely very interesting in the field of 3D-printing and it is an important step forward in finding new technologies for 3D printing. For this reason I would suggest to accept this paper in Nature Communication in the present form without any modifications.

Thank you so much to this reviewer for spending valuable time on the manuscript.

We were encouraged by the comments on NIR-DIW method to provide a new technology for 3D printing and a novel proposal for achieving freestanding structures. It is a great honor to get comments from this reviewer who spent her/his valuable time to work on our materials submitted.

>**Reviewer 3:** Zhu et al. presented the fabrication of three-dimensional architecture enabled by up-conversion nanoparticles (UCNPs), which utilize near-infrared light for the generation of free radicals to initiate the radical polymerization of acrylate monomers and crosslinkers. The introduction of UCNPs greatly reduces the energy required for the crosslinking of a network and increases the light penetration length for a more holistic structure. The authors demonstrated preliminary results in the manuscript, showing interesting aspects of introducing UCNPs into 3D printing. However, early work has shown that up-conversion materials could be employed in stereolithography for building up three-dimensional objects (J. Méndez-Ramos, et al., J. Mater. Chem. C, 2016, 4, 801—806) and it would be advised for the authors to further clarify the design novelty compared with aforementioned work to distinguish a pure adaptation to DIW. Hence, this paper is not novel enough to warrant publication in Nat. Commun. And the reviewer would like to suggest the authors submit this paper somewhere else.

This reviewer's comments helped to improve the quality of the manuscript. The 3D printing strategy presented in the manuscript (DIW, direct ink writing) is

different from early work (J. Me´ndez-Ramos, et al., J. Mater. Chem. C, 2016, 4, 801—806). By applying NIR laser to cure the filaments extruded from varied nozzles, it is possible to realize rapid curing of thick filament (4.00 mm nozzle investigated in this manuscript) which could promise a faster printing process by reducing the time spent on the inner bulk area. Absolutely, the strategy of stereolithography is a genius solution for novel cost-effective luminescent SLA technology comparing to two-photon polymerization (2PP). Nevertheless, the NIR-DIW is aiming for rapid curing of thick filaments or multi-colored or multi-component filaments across different scale levels, where the present photopolymerization-based method (DLP/SLA/DIW with UV region lights) could hardly achieve, and offering a proposal of new method to obtain free-standing structures or multi-color structures. Thank you to this reviewer who spent her/his valuable time to work on our materials submitted.

>1: The manuscript is entitled “3D Printing of Multi-scalable Structures via High Penetration Near-Infrared Photopolymerization” but the concept of “multi-scale” was not addressed in the manuscript, nor did this reviewer trace experiments relating to multi-scale structure. It would be better if the authors would clarify or revise accordingly. This reviewer would like to suggest the authors listing the scales of existing UV-assisted DIW for the purpose to compare.

We revised the manuscript and the concept of “multi-scale” was addressed in

the manuscript. We employed several nozzles with smaller diameter to achieve smaller scale of filaments. Filaments ranged from 17 μm (Figure S3) to 4 mm (Figure 3) had been successfully obtained by the NIR-DIW method. And the similar structure could be achieved in different scale from millimeters to centimeters as shown in Figure 5c.

>2: The schematics in Figure 1 were incorrect topologically since acrylate is not monovalent and the authors should revise this.

We revised the schematics in Figure 1 and corrected the spelling errors. Thanks again for spending time pointing out the mistakes in the manuscript.

>3: The conversion of the vinyl groups was calculated using ATR-FTIR. However, it has been known that IR spectra is more a qualitative rather than a quantitative analysis. This reviewer is concerned about how accurate the conversion obtained by the authors is since all the conversions are presented with three significant numbers.

It was a challenge to exactly figure out what chemically happened during the curing process while the mechanical characteristics could be measured by a real-time rheometer incessantly. In photopolymerization, photo-DSC and photo-FTIR are always employed to investigate the rapid photo-reaction/crosslinking thermodynamically or kinetically. We referred the

measurement of real time NIR/MIR-photorheology proposed by C. Gorsche (Real Time-NIR/MIR-Photorheology: A Versatile Tool for the in Situ Characterization of Photopolymerization Reactions. *Anal. Chem.* 2017, 89, 4958–4968) which could obtain the conversion of the vinyl groups and modulus with the same sample at one time. Besides, we contacted Thermal Fisher who manufactured the set of IR spectrometer and rheometer about the accuracy of the data. For instance, we added error bars in the figures and revised the significant numbers.

>4. The UCNPs in the manuscript, even though promising in near-infrared polymerization, were not provided with any chemical structure, such as the chemical composition, doping elements, size, PDI, and so on. It would be better if the authors could further clarify all the essential information.

We revised the manuscript and added the detailed information of UCNP such as chemical composition, size, PDI and TEM image (Figure S1). The material of UCNP we used was NaYF₄:Yb,M with mean diameter of 1385 nm and 0.396 for polydispersity since the UCNP was commercially available. We are sure the resolution of NIR-DIW will be improved with the development in UCNP or other up-conversion materials.

>5. The freestanding 3D-printed objects in Figure 4, in the opinions of this reviewer, are far from controllable if not intentionally designed that way. This reviewer is concerned

about how precise this technique developed by the authors could be if the ink cannot be solidified. To demonstrate the 3D printability, more 3D printed monolith should be demonstrated other than spiral structure.

We revised the manuscript and added more 3D printed monolith structures in figure 5.

>6. There are numerous grammatical errors. The authors are suggested to proofread more carefully to polish their manuscript significantly. Some (incomplete) examples are shown below.

We revised the manuscript and corrected the spelling errors as well as grammatical errors. Thanks again for spending time pointing out the mistakes in the manuscript.

>7. From this paper, it is unclear to this reviewer how the pigments were integrated into the system and whether the printer colors were robust against different conditions.

The pigments were added during the weighing of ink preparation and integrated into the system by paste mixer (PM300V, TRILOS) at 1200 rpm for 15 min as stated in supplementary information. And we have tested the robustness of the colors in various conditions (heat, humidity and acidic solution).

For thermal shock test, two sets of the sample underwent 10 circles from -20 °C to 150 °C. Afterwards, constant humidity test was performed for half of the structures where the relative humidity was remained 100% at 50 °C for 72 h. Another set of samples was immersed in acidic solutions (1 M HCl solution) separately for 120 h. The optical image of the changes is attached in the following figure.

>8. This reviewer suggests some mechanical measurements on the printed structures to exhibit the robustness of the printed objects.

This is absolutely a good idea to measure the mechanical properties of the printed objects. We have printed structures with pigment-free ink and taken tensile test in which Young's modulus of the sample was 585 MPa. Additionally, we also printed the same structures with colored inks (yellow/red/blue/white, Figure 5a) and there was little difference between pigment-free sample or

exogenously colored sample thanks to the penetration of NIR light.